# Adsorption Behaviour of Reactive Blue 194 on Raw Ramie Yarn in Palm Oil and Water Media

**DOI:** 10.3390/ma15217818

**Published:** 2022-11-05

**Authors:** Lina Lin, Le Li, Lexin Xiao, Cong Zhang, Xueqing Li, Md. Nahid Pervez, Youqing Zhang, Md. Nuruzzaman, Md. Ibrahim H. Mondal, Yingjie Cai, Vincenzo Naddeo

**Affiliations:** 1Hubei Provincial Engineering Laboratory for Clean Production and High Value Utilization of Bio-Based Textile Materials, Wuhan Textile University, Wuhan 430200, China; 2Hubei Key Laboratory of Biomass Fibers and Eco-Dyeing & Finishing, Wuhan Textile University, Wuhan 430200, China; 3Sanitary Environmental Engineering Division (SEED), Department of Civil Engineering, University of Salerno, 84084 Fisciano, Italy; 4Guangzhou Jacky Textile & Technology Co., Ltd., Guangzhou 511338, China; 5Polymer and Textile Research Laboratory, Department of Applied Chemistry and Chemical Engineering, University of Rajshahi, Rajshahi 6205, Bangladesh; 6Institute of Mining, Mineralogy and Metallurgy (IMMM), Bangladesh Council of Scientific and Industrial Research (BCSIR), Khanjanpur 5900, Bangladesh

**Keywords:** sustainable dyeing, grass cloth, adsorption, kinetics, isotherm

## Abstract

As an edible oil, palm oil is also safe and reliable in dyeing, and the residual palm oil after dyeing can be recycled and used continuously, which is green and environmentally friendly and has great research prospects. In this research, raw ramie yarn, used for traditional grass cloth, was dyed in a palm oil medium using Reactive Blue 194. Studying the adsorption and diffusion behaviour in the dyeing process is necessary. Additionally, the kinetics and isotherm model of dyeing raw ramie yarn with Reactive Blue 194 in palm oil is studied, and the adsorption behaviour between them is discussed. For a better understanding, the raw ramie yarn dyeing adsorption behaviour was also carried out in a water medium. It was found that the dyeing rates in palm oil are distinctly faster than in water. Kinetics data suggested that the pseudo-second-order model fitted for both dyeing mediums (palm oil and water) of the adsorption of the Reactive Blue 194 dye onto raw ramie yarn. Afterward, the adsorption isotherms’ results denote that the Langmuir model was suitable for palm oil dyeing medium while the Freundlich model was suited for water medium. Overall, this study has demonstrated that raw ramie yarn dyeing in a palm oil medium could be a sustainable colouration route for textile fibres with a greater dye exhaustion percentage.

## 1. Introduction

Traditional grass cloth, also named Xiabu, refers to the woven fabric produced from raw ramie yarns by handwork (Figure 1), including the manual preparation of raw ramie yarns. Traditional grass cloth differs from modern grass cloth, whose manufacturing process is finished by modern textile machines. Traditional grass cloth has special significance in Chinese textile history as it has existed for more than 6000 years [1]. It is rare to use traditional grass cloth as garment fabric now, but it is commonly used to produce all kinds of decorative items [2,3]. In 2008, the manufacturing process of traditional grass cloth was recorded in the National Intangible Cultural Heritage List, established by the government of China. To maintain and develop the manufacturing technics of traditional grass cloth, the utilization of colourful raw ramie yarns to produce a special pattern of traditional grass cloth is a potential option.

Raw ramie yarns are prepared by tearing the ramie phloem components into small yarns and drying them, and their main constituents include cellulosic fibres and gummy materials [4]. The gummy materials bind each cellulosic fibre [5], resulting in a hard and rough surface with a ditch-like cavity, which contributes a special and unique style to traditional grass cloth. The molecular structures of the gummy components [6] (hemicelluloses, pectin, and lignin) are similar to cellulose, which have free hydroxyl groups and hydrophilic property [7]. Therefore, reactive dye is beneficial for the colouration of raw ramie yarn because it not only offers varieties of brilliant colour, but also gives excellent washfastness by covalently bonding [8,9,10]. Generally, reactive dyes [8,11] of the dichlorotriazinyl group, vinyl sulphone group, or monochlorotriazinyl group are mainly applied for cellulosic fibre dyeing. Dichlorotriazinyl reactive dye can be fixed with cellulosic fibre at low temperatures (30–50 °C) and is mostly used in pad-bath dyeing. However, the residual monochlorotriazinyl group is potentially hydrolysed, and it then generates acidic conditions to accelerate the hydrolyses of a covalent bond between the reactive dye and fibre. Thus, the dichlorotriazinyl reactive dye is gradually abandoned. Vinyl sulphone reactive dye is fixed at medium temperature (50–70 °C), but the stability of the covalent bond between reactive dye and fibre is the poorest among these three reactive dyes. Thereby, reactive dye with a sole vinyl sulphone group is rarely used. For monochlorotriazinly reactive dye, although the covalent bond is the most stable, the required fixation temperature is high (80–100 °C). Thus, the monochlorotriazinyl reactive dye is widely used for printing. Therefore, in the reactive dye category, bi-functional reactive dye includes one monochlorotriazinyl reactive group and one vinyl sulphone reactive group, widely applied in the dyeing of cellulosic fibre because of the medium fixation temperature and the stable covalent bond; for example, Reactive Blue 194 (Figure 2). However, the problems of high water consumption and water pollution, as well as the fact that the dye utilization rate concerning reactive dyeing is unsatisfactory, make it urgent to find a method to solve these problems [12,13]. To reduce water consumption, salt, and alkali usage in reactive dyeing, the dyeing process using vegetable oil as a medium is gradually being tested and approved by researchers [14,15,16,17], which is a cleaner dyeing production and is effective in sorting environmental pollution problems out. However, after reactive dyeing in cottonseed oil, the transparent cottonseed oil became muddy, especially for the dyeing at high pH and high temperature, which was caused by the transformation of the saturated fatty acids from unsaturated ones. This transformation challenges the recycling and reusing of cottonseed oil.

Palm oil is one of the vegetable oils extracted from palm trees. The high yield and low production cost of palm oil make it a competitive vegetable oil [18]. Palm oil has high heat and oxidative stabilities, and it is widely used for food frying [19]. Thus, it is not harmful to humans. Since palm oil has fewer unsaturated fatty acids than other liquid vegetable oils [20], it has resistance to high temperature and high pH conditions. Therefore, using palm oil as a medium in the reactive dyeing of cellulosic fibre can avoid the muddy problem of cottonseed oil after reactive dyeing.

In our previous paper [21], traditional grass cloth dyeing with reactive dye in palm oil medium exhibits a good dyeing performance, including a high dye exhaustion percentage (95.4%) and acceptable dye fixation rate (50.6%). Reactive dyeing in palm oil has some advantages as compared to in water [22]. First, palm oil dyeing is a waterless dyeing process, so it decreases water consumption. Second, the elimination of electrolyte addition in dyeing reduces chemical consumption. Third, the solid reactive dyes in palm oil are exhausted fast and large in the wet substance. Forth, the dispersion of solid reactive dyes in palm oil medium avoids the problem of reactive dye hydrolysis in the dye bath during dyeing; thus, it is effective in the promotion of total dye fixation and eases the reuse of residual dye baths. Finally, it is easy to recycle palm oil, which can be collected by static separation of residual palm oil dye bath with water addition. It is worth noting that after the dyeing, high efficient soaping auxiliary is essential to wash off the palm oil adsorbed in the dyed substance, as well as the unfixed reactive dye. The washed water containing a little palm oil is discharged to the wastewater treatment station. However, palm oil is environmentally friendly and biodegradable [23], so aquatic creatures are unharmed when it is in the water.

The cleaner production of raw ramie yarn dyeing in palm oil medium is framed. The present work is the first to discover the dye adsorption behaviours, including kinetics and isotherm models, in the dyeing of raw ramie yarn in a palm oil medium compared to that in aqueous dyeing. However, the dye fixation treatment was not considered because it prompts further dye exhaustion, i.e., second exhaustion, and consequently interferes with the analysis of the dyeing kinetics and isotherm models. These investigations are beneficial to better understanding the palm oil dyeing process.

## 2. Materials and Methods

### 2.1. Materials

Raw ramie yarn was purchased from the local market. Palm oil was obtained from Feidong Jufeng Grain and Oil Food Factory. Reactive Blue 194 was purchased from Shanghai Jiaying Chemical Company (Shanghai, China).

### 2.2. Dyeing Process

In palm oil medium dyeing, before dyeing, the raw ramie yarn (2 g) was wetted by dipping in water and then padding twice with a 140% pick-up rate. For the kinetic dyeing analysis, the wet raw ramie yarn was dyed with Reactive Blue 194 (0.06 g) in palm oil (40 mL) at a variety of dyeing temperatures from 50–80 °C for 10–360 min with a swing dyeing machine (Foshan Ronggui Huibao Dyeing and Finishing Machinery Factory, Foshan, China). For the thermodynamic dyeing analysis, the wet ramie yarn was dyed with a variety of Reactive Blue 194 dye mass from 0.004 to 0.08 g in palm oil (40 mL) at various dyeing temperatures from 50–80 °C for 360 min with the swing dyeing machine. The dyeing profile is shown in Figure 3a.

In water medium dyeing, the raw ramie yarn (2 g) was dyed with Reactive Blue 194 (0.06 g) in water (40 mL) with 40 g L^−1^ of NaCl at a variety of dyeing temperatures from 40–70 °C for 10–360 min using a swing dyeing machine (Foshan Ronggui Huibao Dyeing and Finishing Machinery Factory, Foshan, China). For the thermodynamic dyeing analysis, the raw ramie yarn was dyed with a variety of Reactive Blue 194 dye mass from 0.004 to 0.08 g in water (40 mL) with 40 g L^−1^ of NaCl at a variety of dyeing temperatures from 40–70 °C for 360 min using the swing dyeing machine. The dyeing profile is shown in Figure 3b.

### 2.3. Dye Exhaustion Percentage

After palm oil dyeing, the excessive dye solution in the dyed sample was removed by hand squeezing and collected with the residual palm oil dye solution. Distilled water (100 mL) was added to the residual palm oil dye solution, followed by a stirring to completely dissolve the reactive dye in the water. Then the aqueous dye solution was separated from the palm oil medium by static separation at 30 °C until the top layer (palm oil) displayed clean (about 3 days). The bottom layer, i.e., aqueous dye solution, was collected to detect its light absorbance with a UV-Vis spectrophotometer (Cary 300, Agilent Technologies, Melbourne, Australia) at its max wavelength (600 nm).

The dye exhaustion percentage (E%) of palm oil dyeing or aqueous dyeing was calculated according to Equation (1). A_0_ and A_1_ refer to the light absorbance of the aqueous dye solution before and after dyeing, respectively [24,25].
(1)$$E\%=\frac{A_{0}-A_{1}}{A_{0}}\times 100\%$$

### 2.4. X-ray Diffraction Analysis

The X-ray diffraction (XRD) analysis of the raw ramie yarns scissored into powders was carried out with an X-ray diffractometer (Rigaku Ultima III, Tokyo, Japan). The XRD pattern was analysed and deconvoluted using the FitYK 1.3.1 software to obtain the characteristic peaks. The crystalline index (CI) of the sample was calculated from the areas of crystalline (I_c_) and amorphous (I_a_) regions by Equation (2) [26].
(2)$$\mathrm{CI}=\frac{I_{C}}{I_{C}+{\;I}_{A}}\times 100$$

## 3. Results and Discussion

### 3.1. Dye Adsorption Behaviour

In reactive dyeing of cellulosic fibre, alkali is essential to fix the adsorbed dye with fibre, which fixation treatment prompts further dye exhaustion, i.e., second exhaustion. To evade interference in the analysis of the dyeing kinetics and isotherm models from the second exhaustion, alkali addition was not considered in the present work.

The dye adsorption behaviours of Reactive Blue 194 in the dyeing of raw ramie yarn in palm oil and water media at different temperatures are shown in Figure 4. In the palm oil medium dyeing (Figure 4a), it is obvious that the dye was exhausted quickly in raw ramie yarn in 50 min; subsequently, the dye adsorption of dyeings increased gradually and tended to be in equilibrium. While in the water medium dyeing (Figure 4b), the dye was adsorbed fast in the first 180 min of the dyeing period and then slightly slow in the later dyeing period. In comparing the dye adsorption behaviour between the dyeings in palm oil and water media, the dye uptake rates in palm oil are distinctly faster than in water. The E% values of dyeings in palm oil at 50, 60, 70, and 80 °C for 50 min were 47.5, 49.3, 51.9, and 52.8%, respectively; whereas the E% values of aqueous dyeing at 40, 50, 60, and 70 °C for 180 min were 32.3, 34.5, 38.2, and 43.1%, respectively.

During the palm oil dyeing, the aqueous soluble Reactive Blue 194 dye in the palm oil was adsorbed in the wet raw ramie yarn at once, and the dye preferred staying in the wet substance rather than transferring into the palm oil. Also, the water adsorbed in the wet raw ramie yarn was repulsed to transfer to the palm oil dye bath because the palm oil medium is hydrophobic. Thus, this dye adsorption was irreversible, i.e., the dye only transferred from the palm oil dye bath to the wet raw ramie yarn [17]. Therefore, in the first 50 min dyeing, the dye was dramatically adsorbed on the wet raw ramie yarn. During the aqueous dyeing, the main components of raw ramie yarn, as well as the Reactive Blue 194, became negative substances caused by ionization. Thereby, a repulsing force [27] between the raw ramie yarn and the dye declined the dye adsorption amount and the dye adsorption rate. Adding salt (NaCl, 40 g L^−1^) in the aqueous dyeing reduced the repulsing force, so the dye exhaustion was promoted. Whereas, in palm oil dyeing, the solid reactive dye was adsorbed first in water content in the wet raw ramie yarn owing to the hydrophilic property of reactive dye. Thus, salt is not essential in palm oil dyeing. In addition, in aqueous dyeing, the dye adsorbed in raw ramie yarn was desorbed and transferred to an aqueous dye bath, i.e., the dye adsorption process was reversible. Therefore, compared to palm oil dyeing, the adsorbed dye amount and the dyeing rate in aqueous dyeing was lower and slower.

When the palm oil dyeing extended to 360 min, the E% of dyeings at 50, 60, 70, and 80 °C only increased to 58.7, 60.0, 60.4, and 61.3%, respectively. While extending the aqueous dyeing time to 360 min from 180 min, the E% of dyeings at 40, 50, 60, and 70 °C increased to 35.4, 37.2, 42.3, and 46.4%, respectively. The increment of E% of aqueous dyeing was higher than that of palm oil dyeing. The amount of dye adsorption in raw ramie yarn in 180 min aqueous dyeing was low, which means that there were many dyes in the dyebath. Meanwhile, adding NaCl (40 g L^−1^) assisted dye adsorption. Furthermore, the aqueous dyeing process is reversible dye adsorption. Thus, the dye adsorption was gradual and increased in the later dyeing period, in contrast to the dye adsorption in palm oil dyeing. However, for 360 min dyeing, the E% of palm oil dyeing was higher than that of aqueous dyeing.

The dyeing temperature supplies energy for distributing the dye in the dye bath [27], increasing dye transfer from the dye bath to raw ramie fibre. Thus, the higher the dyeing temperature, the higher the dye adsorption, identified by the palm oil and especially the aqueous dyeing. The E% curves (Figure 4b) of different dyeing temperatures are more distinct in aqueous dyeing, while the E% curves in palm oil dyeing are closer, suggesting that, in palm oil dyeing, the influence of dyeing temperature on dye absorption was not sensitive due to rapid dye exhaustion.

### 3.2. Kinetic Analysis of Dye Adsorption

Analysis of the kinetic characteristics of dye adsorption in dyeing is helpful for the control and optimization of the dyeing process. The pseudo-first-order and pseudo-second-order models were selected to characterize, and the results are shown in Figure 5 and Table 1. The pseudo-first-order [28] and pseudo-second-order kinetic models [29] are worked out in Equations (3) and (4), respectively.
(3)$$\log\left(q_{e}-q_{t}\right)=logq_{e}-\frac{K_{1}}{2.303}\times t$$
(4)$$\frac{t}{q_{t}}=\frac{1}{k_{2}q_{e}^{2}}+\frac{1}{q_{e}}\times t$$
where *q_e_* (mg g^−^^1^) is the amount of dye adsorbed at equilibrium, *q_t_* (mg g^−^^1^) denotes the amount of dye adsorbed at time *t* (min) and *K*_1_ (min^−^^1^) is the rate constant of the pseudo-first-order model, *K*_2_ (g mg^−^^1^ min^−^^1^) is the rate constant of the pseudo-second-order model.

In the palm oil dyeing (Table 1), linear fit correlation coefficients ($R_{2}^{2}$) of the pseudo-second-order kinetic are all greater than 0.99, while that ($R_{1}^{2}$) of the pseudo-first-order kinetic range from 0.8763 to 0.9280, which are lower than those of the pseudo-first-order kinetic model. In addition, the *q_e_*_(cal1)_ of the pseudo-first-order model is much lower than *q_e_*_(exp.)_, while *q_e_*_(cal2)_ of the pseudo-second-order model is close to *q_e_*_(exp.)_. Therefore, the adsorption behaviour of Reactive Blue 194 in palm oil dyeing of raw ramie yarn fits the pseudo-second-order kinetic model. In the aqueous dyeing, the linear fit correlation coefficients of pseudo-first-order ($R_{1}^{2}$) and pseudo-second-order ($R_{2}^{2}$) are excellent and close, but the $R_{2}^{2}$ is little better than $R_{1}^{2}$ since $R_{2}^{2}$ values are all higher than 0.9900. Therefore, the adsorption behaviour of Reactive Blue 194 in aqueous dyeing of raw ramie yarn fits the pseudo-second-order kinetic model as well [30].

The pseudo-second-order model describes the adsorption identified by two adsorption processes; the first adsorption is fast and quick, an equilibrium situation, whereas the second adsorption takes a long time and is obtuse [31,32]. In Figure 4a, the dye adsorption behaviours fit the pseudo-second-order model description. In the aqueous dyeing, after 180 min dyeing, the dye adsorption still substantially increased, although the dye adsorption rate was lower than in 180 min dyeing. Thus, the increment of dye adsorption leads to the fuzzy kinetic model.

### 3.3. Half-Dyeing Time and Dye Diffusion Coefficient

The half-dyeing time (*t*_1/2_) is the time that the dye mass adsorbed in the substance is equal to half of the dye mass adsorbed at equilibrium, i.e., *q_t_* = 1/2 *q_e_*. It can be calculated from the rate constant of the Pseudo-second-order model, *K*_2_, according to Equation (5) [33].
(5)$$t_{1/2}=\frac{1}{K_{2}\times q_{e\left(cal2\right)}}$$

In Table 2, the *t*_1/2_ values of palm oil dyeing and aqueous dyeing were reduced with increased dyeing temperature. The shorter *t*_1/2_ indicates that the dye adsorption is faster, which is confirmed by the E% curves in Figure 4. Comparing the *t*_1/2_ values between the dyeings in palm oil and in water media, the *t*_1/2_ value of palm oil dyeing shows that dye adsorption is faster in palm oil dyeing.

During the dyeing, the dye on the surface of raw ramie yarn diffused into the fibre interior. Thus, the diffusion coefficient was an important value for characterising the dye adsorption behaviours. The dye diffusion coefficient (*D*) of the raw ramie yarn dyeing in palm oil and water media at *t*_1/2_ can be calculated by the simplified Hill equation (Equation (6)) [34], and the results are listed in Table 2.
(6)$$D=6.292\times 10^{-2}\times \frac{r^{2}}{t_{1/2}}$$
where *D* (m^2^ min^−^^1^) is the dye diffusion coefficient at *t*_1/2_; *r* (m) is the radius of raw ramie yarn (68.9 ± 13.9 μm) [7].

The *D* increased from 1.8576 × 10^−^^11^ to 2.3838 × 10^−^^11^ m^2^ min^−^^1^ with an increase in dyeing temperatures from 50 to 80 °C in palm oil dyeing, and increased from 0.4218 × 10^−^^11^ to 0.6046 × 10^−^^11^ m^2^ min^−^^1^ with an increase of dyeing temperatures from 40 to 70 °C in aqueous dyeing. The dye diffusion in palm oil dyeing is 3 to 4 times faster than in aqueous dyeing. The tendency can be explained by the fact that at a higher dyeing temperature, the raw ramie yarn swelled well, and the dye movement in the dye solution and the fibre was accelerated [35].

In addition, the crystallinity of the raw ramie yarn influences the dye diffusion in the fibre’s interior. The XRD curves of the raw ramie yarn dyed at 80 °C for 360 min in a palm oil medium, the one dyed at 70 °C for 360 min in water, and the original one have displayed in Figure 6, and peaks were deconvoluted using the Gaussian peak fitting function. Afterward, the area under the peaks (different colours) was determined to calculate the crystalline index (CI) using Equation (2). The XRD curves of these samples are similar and fit the cellulose I pattern [36], in which intense peaks at 15.0, 16.6, 22.7, and 34.4° correspond to the (1$\bar{1}$0), (110), (200), and (004) lattice planes, respectively. Besides, the crystallinities of these three samples ranged from 70–72, indicating that the morphological structure of raw ramie yarn was not damaged by the dyeing conditions. Therefore, the fast and high dye adsorption behaviour in the palm oil dyeing of raw ramie yarn was mainly due to irreversible adsorption.

### 3.4. Adsorption Isotherms

Langmuir [37] (Equation (7)) and Freundlich [38] (Equation (8)) isotherms are widely used in fitting the dye adsorption in textile dyeing.
(7)$$\frac{1}{{\left[D\right]}_{f}}=\frac{1}{{\left[S\right]}_{f}\times K_{L}}\times \frac{1}{{\left[D\right]}_{s}}+\frac{1}{{\left[S\right]}_{f}}\;$$
(8)$$log\;{\left[D\right]}_{f}=log\;K_{F}+\left(\frac{1}{n}\right)log\;{\left[D\right]}_{s}\;$$
where [*D*]*_f_* (g g^−^^1^) and [*D*]*_s_* (g L^−^^1^) are the dye concentration in the fibre and dye solution at equilibrium time; [*S*]*_f_* (g g^−^^1^) is the Langmuir constant related to adsorption capacity; *K_L_* is the Langmuir equilibrium constant. *K_F_* is the Freundlich equilibrium constant, and n is the constant correlated to the adsorption intensity of the adsorbent.

Langmuir isotherm assumes the adsorption occurs on a homogeneous surface by monolayer adsorption without interaction between adsorbed molecules [39]. In contrast, Freundlich isotherm assumes that adsorption occurs on heterogeneous surfaces by monolayer and multilayer adsorption [40]. Both isotherms can be chemisorption and physisorption. In the case of reactive dyeing, it is hard to clear distinct the sole chemical or physical adsorption since the reactive group of the reactive dye can partly chemically react with the substance to form covalent bonds [41]. The Langmuir and Freundlich isotherms for dyeing raw ramie yarn with Reactive Blue 194 in palm oil medium are shown in Figure 7a,b, and Table 3, and those for dyeing in water medium are shown in Figure 7c,d, and Table 4. The linear relationship of the Langmuir isotherm (Figure 7a,c) and Freundlich isotherm (Figure 7b,d) is high fitness, which average correlation coefficients higher than 0.9200, suggesting that the dyeing of raw ramie yarn in the palm oil medium and water medium both chemical and physical adsorptions existed. In the dyeing of raw ramie yarn in the palm oil medium, the average correlation coefficient of the Freundlich isotherm ($R_{F}^{2}=0.9774$) is higher than that of Langmuir isotherm ($R_{L}^{2}=0.9235$). It hints that the dye adsorption tended to be physical adsorption, more fittable to the Freundlich isotherm [42]. While the dyeing of raw ramie yarn in a water medium exhibited a competition of chemical and physical adsorption because the average correlation coefficients of the Langmuir isotherm ($R_{L}^{2}$) and Freundlich isotherm ($R_{F}^{2}$) are 0.9833 and 0.9714, respectively. In the aqueous dyeings, the addition of 40 g L^−^^1^ of NaCl promoted dye adsorption, which enhanced the potential to form covalent bonds between the dye and raw ramie yarn, i.e., chemical adsorption [43,44].

## 4. Conclusions

This study investigated using palm oil and water mediums for the environmentally friendly sustainable dyeing of grass cloth fibre utilizing commercial Reactive Blue 194 dye. The adsorption behaviour of reactive dye on raw ramie yarn was investigated to better understand the dyeing process of reactive dye in palm oil and water media, which ultimately optimizes the dyeing techniques. The dyeing of grass cloth yarns by reactive blue 194 dyes in palm oil and water medium is in accordance with the quasi-secondary kinetic model. The half-dyeing time becomes smaller by increasing the temperature. In the palm oil dyeing system, the dyeing temperature greatly influences its activation energy. The adsorption isotherm of reactive blue 194 dye on grass cloth yarn in palm oil is more consistent with the Freundlich isotherm adsorption model, while the Langmuir isotherm adsorption model was suitable for water dyeing. The affinity of reactive blue 194 dye for dyeing grass cloth yarn in palm oil increased with the increase of temperature, the trend of dye transfer from the dye solution to the fibre was greater, and the dyeing process was a heat absorption reaction. Besides, it was found that the adsorption rates in palm oil are distinctly faster than in water, indicating that a palm oil dyeing medium could be an alternative to a water dyeing medium with a greater dye exhaustion performance, beneficial for the textile dyeing industry in the future regarding environmental sustainability.

## Figures and Tables

**Figure 1 materials-15-07818-f001:**
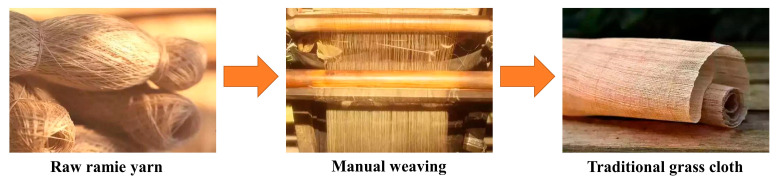
Preparation of traditional grass cloth.

**Figure 2 materials-15-07818-f002:**
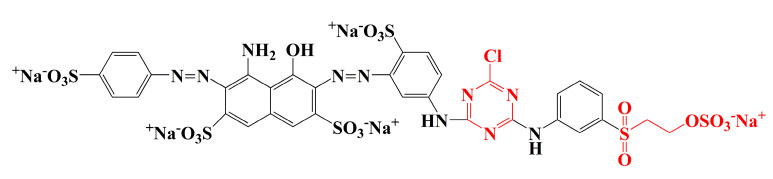
Molecular structure of Reactive Blue 194.

**Figure 3 materials-15-07818-f003:**
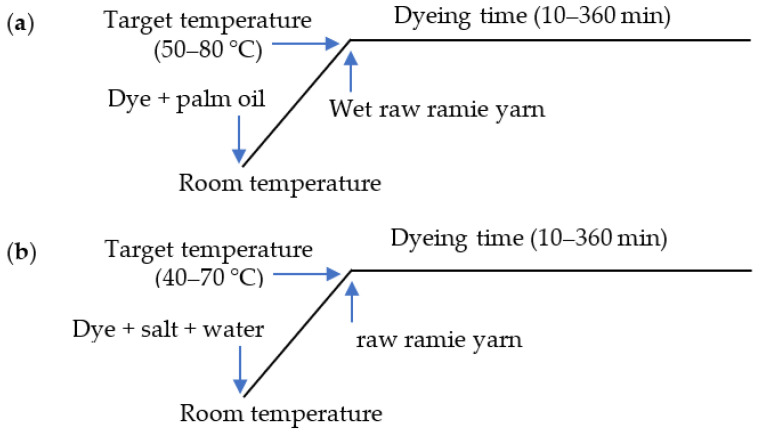
Dyeing profile of raw ramie yarn with Reactive Blue 194 in (**a**) palm oil and (**b**) water media.

**Figure 4 materials-15-07818-f004:**
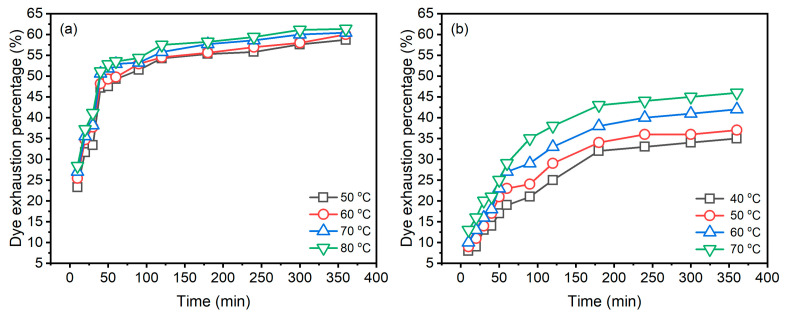
Dye exhaustion of Reactive Blue 194 in the dyeing of raw ramie yarn in (**a**) palm oil medium and (**b**) water medium.

**Figure 5 materials-15-07818-f005:**
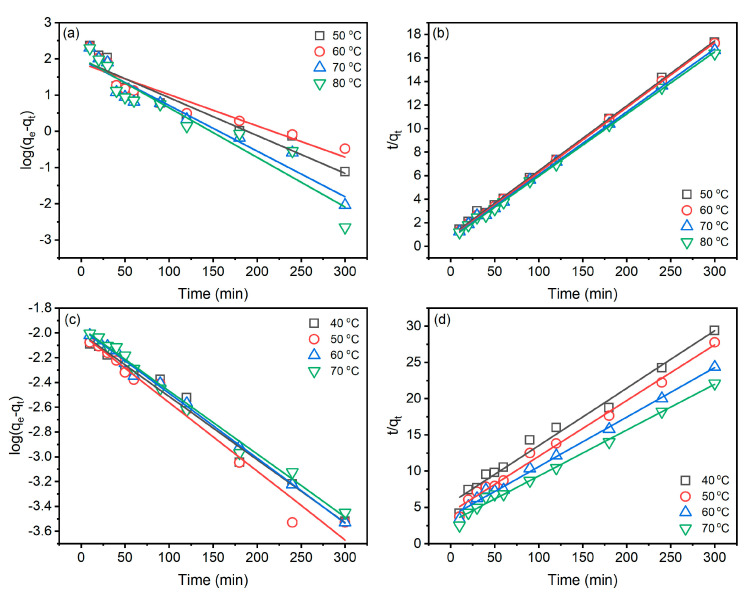
Kinetic characteristics of Reactive Blue 194 dye adsorption in the dyeing of raw ramie yarn in palm oil medium: (**a**) pseudo-first-order and (**b**) pseudo-second-order and in water medium: (**c**) pseudo-first-order and (**d**) pseudo-second-order.

**Figure 6 materials-15-07818-f006:**
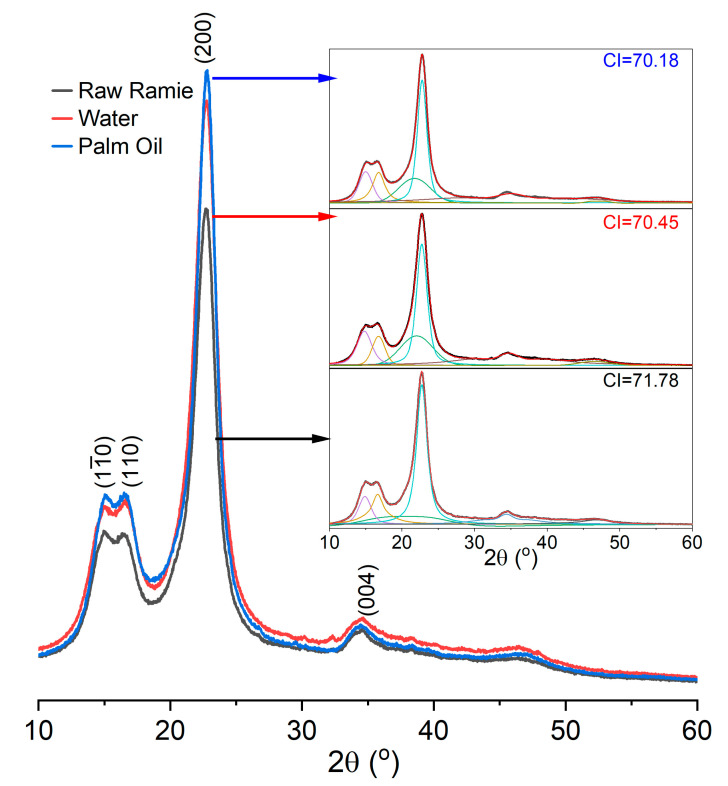
XRD patterns of original raw ramie yarn and raw ramie yarn dyed in the palm oil medium and the water medium, and original raw ramie yarn.

**Figure 7 materials-15-07818-f007:**
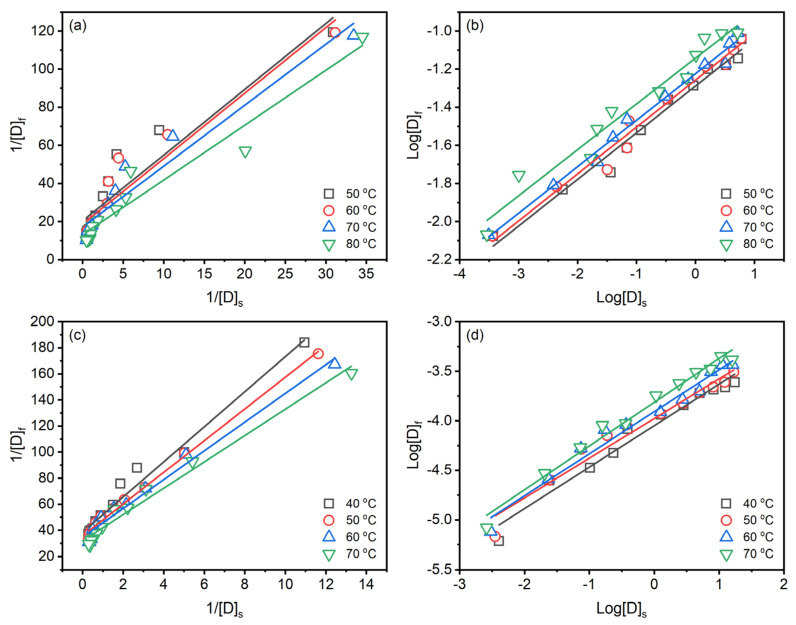
Isotherms of dyeing of raw ramie yarn with Reactive Blue 194 in palm oil medium: (**a**) Langmuir and (**b**) Freundlich, and in water medium: (**c**) Langmuir and (**d**) Freundlich.

**Table 1 materials-15-07818-t001:** Kinetic model of Reactive Blue 194 dyeing raw ramie yarn.

Temp. (°C)	*q_e_*_(exp.)_ (mg g^−1^)	Pseudo-First-Order			Pseudo-Second-Order		
		*K*_1_ (min^−1^)	*q_e_*_(cal1)_ (mg g^−1^)	$R_{1}^{2}$	*K*_2_ (g mg^−1^ min^−1^)	*q_e_*_(cal2)_ (mg g^−1^)	$R_{2}^{2}$
Palm oil medium							
50	17.6098	0.0105	7.2044	0.8982	0.0034	18.1159	0.9986
60	18.0038	0.0086	6.5398	0.8763	0.0041	18.2236	0.9995
70	18.1304	0.0126	7.2109	0.9280	0.0040	18.6916	0.9991
80	18.3978	0.0137	7.5337	0.9116	0.0042	18.9394	0.9994
Water medium							
40	10.4950	0.0054	10.1812	0.9840	0.0011	12.6904	0.9923
50	11.0913	0.0056	9.8969	0.9796	0.0014	12.9702	0.9914
60	12.6045	0.0054	11.2176	0.9956	0.0014	14.7059	0.9926
70	13.6881	0.0051	11.0027	0.9913	0.0013	15.6250	0.9925

**Table 2 materials-15-07818-t002:** Half-dyeing time and dye diffusion coefficient of Reactive Blue 194 dyeing raw ramie yarn.

Temperature (°C)	*t*_1/2_ (min)	*D* × 10^11^ (m^2^ min^−1^)
Palm oil medium		
50	16.08	1.8576
60	13.55	2.2044
70	13.44	2.2224
80	12.53	2.3838
Water medium		
40	70.82	0.4218
50	55.99	0.5335
60	49.75	0.6004
70	49.40	0.6046

**Table 3 materials-15-07818-t003:** Parameters of Langmuir and Freundlich isotherms of dyeing of raw ramie yarn with Reactive Blue 194 in palm oil medium.

Temperature (°C)	50	60	70	80
Langmuir isotherm				
[*S*]*_f_* (g g^−1^)	0.0490	0.0536	0.0584	0.0764
*K_L_*	5.9106	5.4205	5.3417	4.5486
$R_{L}^{2}$	0.8950	0.9147	0.9402	0.9441
	Average: 0.9235			
Freundlich isotherm				
n	4.0811	4.0556	4.0932	4.1559
*K_F_*	0.0515	0.0553	0.0598	0.0722
$R_{F}^{2}$	0.9794	0.9769	0.9888	0.9644
	Average: 0.9774			

**Table 4 materials-15-07818-t004:** Parameters of Langmuir and Freundlich isotherms of dyeing of raw ramie yarn with Reactive Blue 194 in water medium.

Temperature (°C)	40	50	60	70
Langmuir isotherm				
[*S*]*_f_* (mg g^−1^)	0.0259	0.0279	0.0289	0.0314
*K_L_*	2.8742	2.9455	3.1435	3.1543
$R_{L}^{2}$	0.9770	0.9947	0.9824	0.9793
	Average: 0.9833			
Freundlich isotherm				
n	2.3892	2.4932	2.3587	2.2647
*K_F_*	0.0001	0.0001	0.0001	0.0002
$R_{F}^{2}$	0.9688	0.9600	0.9735	0.9833
	Average: 0.9714			

## Data Availability

The datasets generated during the current study are available from the corresponding author on reasonable request (Yingjie Cai, Y.C.).

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
