# Peer review of "Adsorption Behaviour of Reactive Blue 194 on Raw Ramie Yarn in Palm Oil and Water Media"

_materials, 2022, doi:10.3390/ma15217818_

Round 1
Reviewer 1 Report
The current manuscript describes the Adsorption Behaviour of Reactive Blue 194 Dyeing Raw Ramie Yarn in Palm Oil and Water Media. The application of the selected dye is described in a superficial manner. However, there are quite a few major concerns and inconsistencies with respect to the typical reactive dyeing parameters and dye-fiber fixation.
One of the major drawbacks of this manuscript is that the authors have limited the present work to the adsorption (neutral) stage relative to the typical salt phase in aqueous medium. Regarding to the typical reactive dyeing process, do the authors believe that the adsorption phase can promote the typical phases of primary and secondary exhaustion. I would rather ask the authors to complete the dyeing cycle studying the dye fixation under the alkaline condition in Palm Oil. Also, the authors have switched off the application onto the raw fiber not to the scoured or bleached ones. The hydrophobic/hydrophilic chacter of the fabric used should be taken into consideration. This seems questionable for one needs to follow the proposed dyeing system.
I am still not happy with the inserted text in the introduction. While it does mention previous waterless systems of the general type covered in the paper and does give references, the paper must be far more explicit about the typical reactive dyeing parameters such as dye exhaustion and dye-fiber fixation. The introduction must also give far clearer reasoning as to why the authors wanted to revisit the proposed system for reactive dyestuffs during the whole dyeing process so that readers can understand why time was spent on the experiments and why they should actually read the paper.
Author Response
The current manuscript describes the Adsorption Behaviour of Reactive Blue 194 Dyeing Raw Ramie Yarn in Palm Oil and Water Media. The application of the selected dye is described in a superficial manner. However, there are quite a few major concerns and inconsistencies with respect to the typical reactive dyeing parameters and dye-fiber fixation.
- One of the major drawbacks of this manuscript is that the authors have limited the present work to the adsorption (neutral) stage relative to the typical salt phase in aqueous medium. Regarding to the typical reactive dyeing process, do the authors believe that the adsorption phase can promote the typical phases of primary and secondary exhaustion. I would rather ask the authors to complete the dyeing cycle studying the dye fixation under the alkaline condition in Palm Oil.
Response: Thanks for your valued comments. We apologize for not clearly pointing out the reason for selection of dyeing conditions. We have added more information to describe the dye selection in the “Introduction” part as follows:
“Generally, reactive dyes with dichlorotriazinyl group, vinyl sulphone group, or monochlorotriazinyl group are mainly applied for cellulosic fibre dyeing. Dichlorotriazinyl reactive dye can be fixed with cellulosic fibre at low temperatures (30-50oC), and is mostly used in pad-bath dyeing, but the residual monochlorotriazinyl group is potentially hydrolysed and then generates acidic conditions to accelate the hydrolyses of a covalent bond between the reactive dye and fibre. Thus, the dichlorotriazinyl reactive dye is gradually abandoned. Vinyl sulphone reactive dye is fixed at medium temperature (50-70oC), but the stability of the covalent bond between reactive dye and fibre is the poorest among these three types of reactive dyes. Thereby, reactive dye with a sole vinyl sulphone group is rarely used. For monochlorotriazinly reactive dye, although the covalent bond is the most stable, the required fixation temperature is high (80-100oC). Thus, the monochlorotriazinyl reactive dye is widely used for printing.”
We believe that after the addition of alkali, the dye exhaustion will further promote, i.e., secondary exhaustion. In our previous research, we investigated the dye adsorption behaviour in aqueous dyeing of cotton fibre with salt and soda ash, and the dye solution was taken from the starting dyeing (room temperature), i.e., the practical dyeing process. We found that the data does not fit well with the classic kinetic and isotherm models. After discussion and reading similar research papers, the investigation of dye adsorption behavior in the published paper, the substance was immersed in the dye solution at the target temperature rather than at starting temperature (room temperature). Thus, in this manuscript, the raw ramie yarn was immersed in palm oil and aqueous dye solutions at the target temperature.
We did not focus on dye fixation in this manuscript. As the reviewer commented, the addition of alkali to fix reactive dye can occur second exhaustion because the dye adsorption balance was broken, which causes the promotion of reactive dye adsorption in the substance from the dye bath. In the analyse of the dyeing kinetic and isotherm models, the treating conditions should not change. In aqueous dyeing, if the alkali and salt are added to the dyebath, it is not doubted that the reactive dye is easily hydrolysed, which will influence the dye adsorption behaviour, since more negative charges of dye molecules are generated. Another reason for not considering the dye fixation is that the alkali was dissolved in the pre-wet solution in palm oil dyeing. It differs from dyeing in an aqueous solution, in which the alkali is added at the target temperature. Thus, the manner of alkali addition in palm oil and aqueous dyeings is different.
Salt was added in the aqueous dyeing because the salt does not change the dye molecular structure, and it is usually added in the initial dyebath. Without salt addition, the dye exhaustion in aqueous dyeing is very poor. Thus, it can be shown that the dye exhaustion in palm oil dyeing is better.
We really appreciate these valued comments, and in the next manuscript, we will do the practical dyeing of kinetics and isotherm because it is more useful to optimize the dyeing conditions.
We have modified the introduction part, and some sentences were added with red color marks.
- Also, the authors have switched off the application onto the raw fiber not to the scoured or bleached ones. The hydrophobic/hydrophilic chacter of the fabric used should be taken into consideration. This seems questionable for one needs to follow the proposed dyeing system.
Response: The raw ramie fibre constitutes gummy materials and cellulosic fibre. The stiffness performance was contributed by the gummy materials. We believe that scouring and bleaching can improve the dye exhaustion, but it also considerably removes the gummy materials, resulting in decreasing the stiffness property. The hydrophobic/hydrophilic character was reported in our previous published paper, and the description was added to this manuscript with a red colour mark.
- I am still not happy with the inserted text in the introduction. While it does mention previous waterless systems of the general type covered in the paper and does give references, the paper must be far more explicit about the typical reactive dyeing parameters such as dye exhaustion and dye-fibre fixation. The introduction must also give far clearer reasoning as to why the authors wanted to revisit the proposed system for reactive dyestuffs during the whole dyeing process so that readers can understand why time was spent on the experiments and why they should actually read the paper.
Response: Thanks for your suggestions. As explained above, we did not consider the dye-fibre fixation in this manuscript, but we will do so in the next work. The reason for not considering dye fixation was given in the introduction part. Besides, the advantages of using palm oil as a dyeing medium were described in the introduction part, which is the reason for selecting palm oil as a dyeing medium.
Reviewer 2 Report
1- In section 2.2, the dyeing process is not explained clearly. Is the fixation step by addition of alkali is eliminated? If not eliminate, how was done?
2- The souring after dyeing is not described.
3- How the reactive dye was dispersed in the palm oil? Was any dispersing agent used?
4- The discussion about figure 3 should be improved.
5- An analysis of the ingredients of palm oil and comparison with cottonseed oil or others oils used in the previous studies will help showing the reason for using the palm oil in this study.
Author Response
- In section 2.2, the dyeing process is not explained clearly. Is the fixation step by addition of alkali is eliminated? If not eliminate, how was done?
Response: Thanks for your comments. The profile of the dyeing process (Figure 3) was added in section 2.2 to clearly describe the dyeing experiment. Besides, we did not consider the dye fixation because the addition of alkali at the target temperature will cause second exhaustion. While in the kinetic and isotherms analyse, the dyeing conditions should not change during the dyeing. If the addition of alkali in the initial dye solution at the target temperature, the reactive dye will be hydrolysed, and its molecular structure will be changed as well. These changes will influence the analyse of the kinetic and isotherms.
- The souring after dyeing is not described.
Response: Thanks for your comment. Since we just focus on the investigation of the dye adsorption behaviours, the souring treatment can be eliminated.
- How the reactive dye was dispersed in the palm oil? Was any dispersing agent used?
Response: Thanks for your comments. The solid reactive dye was not dissolved in palm oil, and during the dyeing, the reactive dye particles were surrounded by the palm oil. We have removed the “dispersed” word (2 positions) from the manuscript.
- The discussion about figure 3 should be improved.
Response: thanks for your suggestion, we have improved the discussion about figure 3, in which modifications were marked with red color.
- An analysis of the ingredients of palm oil and comparison with cottonseed oil or others oils used in the previous studies will help show the reason for using palm oil in this study.
Response: Thanks for your comment. We totally agree with your suggestion of analyse of the ingredients of palm oil and other plant oil. That is why in the “Introduction” part, we pointed out the reason for using palm oil as a dyeing medium. However, we checked the ingredients of palm oil products, and there is not about the percentage of unsaturated fatty acid information, and we cannot find the unsaturated fatty acid in the cottonseed oil that was used in the published paper. We hope this explanation meet your requirement.
Reviewer 3 Report
The manuscript of Lin et al. “Adsorption Behaviour of Reactive Blue 194 Dyeing Raw Ramie 2 Yarn in Palm Oil and Water Media” is supposed to be focused on the adsorption studies, in case of “the palm oil dyeing process” compared to “water medium dyeing”.
It is not clear from the manuscript what is the novelty of this work and whether it is of international or only local importance?!
There are several articles on the same topic in the literature, but in which the adsorption process is much better studied and presented.
An article was recently published in which Cong Zhang is a co-author https://doi.org/10.1016/j.indcrop.2021.114315 which addresses the same topic and which also includes the dye “Reactive Blue 194”. It is not “plagiarism”??
If the article is about the adsorption process, then it must be studied according to the parameters that influence the process and also the mechanism of the process must be highlighted.
Lack of study of adsorption parameters: initial concentration, dosage, solution pH, etc.
In order to elucidate the possible mechanism of dye removal, the kinetic studies must be thorough.
The data obtained using intraparticle diffusion model must be analysed and presented, in order to explain the diffusion mechanism. I believe that the values of intraparticle diffusion rate constant (ki) and the effect of boundary and the layer thickness (l) will give important information in order to explain the diffusion mechanism of adsorption.
In order to elucidate the possible mechanism of dye removal, the equilibrium studies must be improved. The authors investigated only the linear regression adsorption models. By using the non-linear forms, the analysis of experimental data is more precise. The adsorption isotherm model study discussion should be extended; there are more isotherms to investigate: Redlich-Peterson, Sips, Dubinin-Radushkevich, Temkin, etc in order to elucidate the adsorption mechanism. Based on the best fitted isotherm model, the adsorption mechanism should be discussed.
It seems that the number of authors (11) is overestimated, it is not justified for the "work" presented in the work. More, at “Author Contributions”:
- five people for "methodology"? The presented adsorption studies require only one person to perform
- for “formal analysis and investigation” eight people? Only "X-ray Diffraction Analysis" and "influence of temperature" were performed!!
- four people to “writing –original draft preparation” 11 pages (3/4 written), of which 4 are only figures/tables???
I made the observations in the .pdf form of the paper, and I hope that this review is helpful to the authors.
I appreciate that in these conditions the manuscript does not meet the standards of the Materials journal.

Author Response
The manuscript of Lin et al. “Adsorption Behaviour of Reactive Blue 194 Dyeing Raw Ramie 2 Yarn in Palm Oil and Water Media” is supposed to be focused on the adsorption studies, in case of “the palm oil dyeing process” compared to “water medium dyeing”.
- It is not clear from the manuscript what is the novelty of this work and whether it is of international or only local importance?!
Response: Thanks for your comments. We modified the last paragraph of the “Introduction” part with red color mark to clearly point out the aim of this work. Besides, ramie is planted in many Asian countries, so, about this work, I think it is of international importance.
- There are several articles on the same topic in the literature, but in which the adsorption process is much better studied and presented.
An article was recently published in which Cong Zhang is a co-author https://doi.org/10.1016/j.indcrop.2021.114315 which addresses the same topic and which also includes the dye “Reactive Blue 194”. It is not “plagiarism”??
Response: Thanks for your comments. In the mentioned published paper, Yingjie Cai and Cong Zhang are the co-authors. That manuscript reported the dyeing of raw ramie yarn with 3 reactive dyes (including reactive blue 194) in a water medium, and the best dyeing performance is for Reactive Orange 5. Subsequently, the detail of dye adsorption and diffusion was only discussed with Orange 5. While this manuscript focuses on the dyeing of raw ramie yarn with reactive dye 194 in a palm oil medium. The selection of Reactive Blue 194 for palm oil dyeing is suggested by a dyeing plant. Generally, reactive dyes with dichlorotriazinyl group, vinyl sulphone group, or monochlorotriazinyl group are mainly applied for cellulosic fibre dyeing. Dichlorotriazinyl reactive dye can be fixed with cellulosic fibre at low temperature (30-50oC), and is mostly used in pad-bath dyeing, but the residual monochlorotriazinyl group is potentially hydrolysed and then generates acidic conditions to accelerate the hydrolyses of a covalent bond between the reactive dye and fibre. Thus, the dichlorotriazinyl reactive dye is gradually abandoned. Vinyl sulphone reactive dye is fixed at medium temperature (50-70oC), but the stability of the covalent bond between reactive dye and fibre is the poorest among these three types of reactive dyes. Thereby, reactive dye with a sole vinyl sulphone group is rarely used. For monochlorotriazinly reactive dye, although the covalent bond is the most stable, the required fixation temperature is high (80-100oC). Thus, the monochlorotriazinyl reactive dye is widely used for printing. Therefore, Reactive blue 194 was suggested to investigate the dyeing of raw ramie yarn in palm oil medium. Since we have done the dyeing with Reactive blue 194 in a water medium, thus, we used these data to compare the dyeing in a palm oil medium. Meanwhile, we did the analyses of kinetics and isotherm model of the water dyeing, which are not discussed in the mentioned published paper. We have added a notice in the present manuscript to claim that the data of reactive blue 194 dyeing was adapted from the published paper with the reference in Figure 4 title. Thus, we do not think it is not plagiarism.
- If the article is about the adsorption process, then it must be studied according to the parameters that influence the process and also the mechanism of the process must be highlighted.
Lack of study of adsorption parameters: initial concentration, dosage, solution pH, etc.
Response: Thanks for your suggestions. Actually, the dyeing process can be considered as the adsorption process, but it is a bit different from the adsorbent for dye removal. In dyeing, the dyed substance's colour must match the sample, i.e., the dye mass usage is fixed. Therefore, in this manuscript, we did not change the initial dye concentration. Besides, in reactive dyeing, the dye adsorption process should be in neutral conditions because at alkali conditions, the reactive group of reactive dye is hydrolysed quickly and loses its reactivity, resulting in a reduction of dye fixation rate, thus we kept the solution pH at the neutral conditions.
- In order to elucidate the possible mechanism of dye removal, the kinetic studies must be thorough.
The data obtained using intraparticle diffusion model must be analysed and presented, in order to explain the diffusion mechanism. I believe that the values of intraparticle diffusion rate constant (ki) and the effect of boundary and the layer thickness (l) will give important information in order to explain the diffusion mechanism of adsorption.
Response: Thanks for your suggestion. The present work only focuses on dyeing, not dye removal. And in textile dyeing, according to the huge of similar dyeing papers, the pseudo-first-order, and pseudo-second-order kinetic models are frequently used; therefore, we used both models for our manuscript. in addition, in our previous published paper (https://doi.org/10.1016/j.indcrop.2021.114315), the intraparticle diffusion model was used, but the fitness of intraparticle was lower than the pseudo-second-order kinetic model. Besides, the intraparticle diffusion model in our previous paper was analysed and discussed by the co-author, prof. Md. Ikram Ul Hoque, while in this manuscript, we did not ask for his help. Thus, we did not include the intraparticle diffusion model in this manuscript.
- In order to elucidate the possible mechanism of dye removal, the equilibrium studies must be improved. The authors investigated only the linear regression adsorption models. By using the non-linear forms, the analysis of experimental data is more precise. The adsorption isotherm model study discussion should be extended; there are more isotherms to investigate: Redlich-Peterson, Sips, Dubinin-Radushkevich, Temkin, etc in order to elucidate the adsorption mechanism.Based on the best fitted isotherm model, the adsorption mechanism should be discussed.
Response: Thanks for your valued comments. We agreed that to extend the equilibrium period, the analysis of experimental data is more precise. However, in fibre dyeing with reactive dye, the time for dye fixation is not more than 60 min. In this manuscript, we extended the dyeing time up to 360 min. that is the reason for the selection of 360 min dyeing time. Anyway, we accepted your suggestion and will extend the dyeing time to be longer.
For the isotherms models, we really think about sharing these models. To be honest, we did not know these isotherms. We will learn these models soon and try to use them in our future research. In fibre dyeings, Langmuir, Freundlich, and Nernst equations are known as the classic models, and the Nernst model is identified to fit the dyeing of synthetic fibres with dispersed dye, especially in dispersed dyeing of polyester. Therefore, according to the property of raw ramie yarn, we decided to use only Langmuir and Freundlich equation to analyse the experimental data.
- It seems that the number of authors (11) is overestimated, it is not justified for the "work" presented in the work. More, at “Author Contributions”:
- five people for "methodology"? The presented adsorption studies require only one person to perform
- for “formal analysis and investigation” eight people? Only "X-ray Diffraction Analysis" and "influence of temperature" were performed!!
- four people to “writing –original draft preparation” 11 pages (3/4 written), of which 4 are only figures/tables???
Response: Thank you for your deep concern about our work.
- Yes, we agree with you that the presented adsorption studies require only one person to perform. However, this work was done as a reference for many new scholars who will be professional researchers in the near future, and therefore, we decided to incorporate five people in the methodology part. Besides, replicating and batch studies experimental patterns needed many people, as you know.
- Formal analysis and investigation section, we added 8 people since, after the experimental run, we needed to discuss the scientific parts with our PI and other lab members. Hope you can correlate.
- Yes, we required 4 people for writing –original draft preparation section. After writing, the first author and two other authors (experienced researchers) updated the draft, and then the PI checked before the final submission.
- I made the observations in the .pdf form of the paper, and I hope that this review is helpful to the authors.
Response: We really appreciated your comments on this manuscript. We have corrected and modified the sentences marked by your comments.
- I appreciate that in these conditions the manuscript does not meet the standards of the Materials
Response: We have revised the manuscript according to 3 reviewers’ comments. We hope this revised manuscript can meet your requirements.
Round 2
Reviewer 1 Report
I think the justification for this manuscript has not quite expressed. There are still some more relevant references in the introduction part and additional experiments needed. The authors concluded that the palm oil dyeing medium could be an alternative to water dyeing medium with a greater dye exhaustion performance which is beneficial for future textile dyeing industry in the sense of environmental sustainability - How can the proposed reactive dyeing system be effective for promoting a satisfactory results of the dye-fiber fixation. The authors should include the potential applying the palm oil dyeing medium under the alkaline conditions so as to justify its effect on the targeted reactive dyeing system.
Author Response
Response: Thanks for your comments. We agreed that it is essential to show a good dyeing performance of raw ramie yarn in a palm oil medium, then we can give the conclusion that “the palm oil dyeing medium could be an alternative to water dyeing medium with a greater dye exhaustion performance which is beneficial for future textile dyeing industry in the sense of environmental sustainability”. In our previous papers, we did the dyeings of traditional grass cloth constituted of raw ramie yarn in palm oil medium and water medium, and an orthogonal experimental scheme and Taguchi analysis were applied to investigate the influence of the variables of pH of the solution, fixation temperature, and fixation time on dyeing performance and acquire the optimal dyeing conditions. Therefore, we give a short sentence marked by red colour in the “Introduction” part of the present manuscript to support the conclusion as follows:
“In our previous paper [22], traditional grass cloth dyeing with reactive dye in palm oil medium exhibits a good dyeing performance, included high dye exhastuion percentage (95.4%) and acceptable dye fixation rate (50.6%). Reactive dyeing in palm oil has some advantages compared to that in water [23].”
Besides, we have cited 2 references for the reactive dyes description and 2 references for the advantages of palm oil dyeing in the “Introduction” part.
We hope the revises can meet your requirements.
References:
- Lin, L.; Xiao, L.; Li, L.; Zhang, C.; Pervez, M.N.; Naddeo, V.; Zhang, Y.; Islame, M.S.; Cai, Y.; Hassan, M.M., Sustainable and eco-friendly dyeing of traditional grass cloth with a reactive dye in palm oil medium. RSC Advances Accepted.
- Lin, L.; Jiang, T.; Li, L.; Pervez, M.N.; Zhang, C.; Yan, C.; Cai, Y.; Naddeo, V., Sustainable traditional grass cloth fiber dyeing using the Taguchi L16 (4^4) orthogonal design. Scientific Reports 2022, 12, (1), 13833.
Reviewer 3 Report
The manuscript of Lin et al. “Adsorption Behaviour of Reactive Blue 194 Dyeing Raw Ramie 2 Yarn in Palm Oil and Water Media” was revised.
The authors answered to the main problems. As a result of the authors' response to my comments, the paper could be ACCEPTED,
with the amendment related to the number of authors for this paper. The answer provided is not "scientifically correct": “Yes, we agree with you that the
presented adsorption studies require only one person to perform. However, this work
was done as a reference for many new scholars who will be professional researchers
in the near future, and therefore, we decided to incorporate five people in the methodology part.”
In a scientific paper (published paper), the persons directly involved, who contributed to the results/discussions are indicated/mentioned!
“Formal analysis and investigation section, we added 8 people since, after the experimental run, we needed to discuss the scientific parts with our PI and other lab members.”
Eight people to discuss the results obtained (not many), seems excessive to me.
Author Response
The manuscript of Lin et al. “Adsorption Behaviour of Reactive Blue 194 Dyeing Raw Ramie 2 Yarn in Palm Oil and Water Media” was revised.
The authors answered to the main problems. As a result of the authors' response to my comments, the paper could be ACCEPTED,
Response: Thank you for your positive evaluation of our revised manuscript.
with the amendment related to the number of authors for this paper. The answer provided is not "scientifically correct": “Yes, we agree with you that thepresented adsorption studies require only one person to perform. However, this workwas done as a reference for many new scholars who will be professional researchersin the near future, and therefore, we decided to incorporate five people in the methodology part.”
Response: Thank you for your concern. Please see that the answer has been modified.
“Yes, we agree with you that the presented adsorption studies require only one person to perform. However, doing replicates of the experimental part, we included several researchers; some were not present on time because of health issues during the Covid 19 situation.
In a scientific paper (published paper), the persons directly involved, who contributed to the results/discussions are indicated/mentioned!
“Formal analysis and investigation section, we added 8 people since, after the experimental run, we needed to discuss the scientific parts with our PI and other lab members.”
Eight people to discuss the results obtained (not many), seems excessive to me.
Response: Thank you for your kind concern. Although we discussed this with eight people, we do not think all should be included in this section, therefore we modified this section. Please see that we updated the formal analysis and investigation section with only 3 people who were exactly correlated with this work.